# Cell Death in Coronavirus Infections: Uncovering Its Role during COVID-19

**DOI:** 10.3390/cells10071585

**Published:** 2021-06-23

**Authors:** Annamaria Paolini, Rebecca Borella, Sara De Biasi, Anita Neroni, Marco Mattioli, Domenico Lo Tartaro, Cecilia Simonini, Laura Franceschini, Gerolamo Cicco, Anna Maria Piparo, Andrea Cossarizza, Lara Gibellini

**Affiliations:** Department of Medical and Surgical Sciences for Children and Adults, University of Modena and Reggio Emilia, 41121 Modena, Italy; annamaria.paolini@unimore.it (A.P.); rebeccaborella7993@gmail.com (R.B.); debiasisara@yahoo.it (S.D.B.); anitaneroni@hotmail.it (A.N.); mattioli.marco@gmail.com (M.M.); domenico.lotartaro@gmail.com (D.L.T.); ceciliasimonini24@gmail.com (C.S.); 228164@studenti.unimore.it (L.F.); jerrycicco.1@gmail.com (G.C.); piparoannamaria@gmail.com (A.M.P.); lara.gibellini@unimore.it (L.G.)

**Keywords:** cell death, COVID-19, apoptosis

## Abstract

Cell death mechanisms are crucial to maintain an appropriate environment for the functionality of healthy cells. However, during viral infections, dysregulation of these processes can be present and can participate in the pathogenetic mechanisms of the disease. In this review, we describe some features of severe acute respiratory syndrome coronavirus 2 (SARS-CoV-2), and some immunopathogenic mechanisms characterizing the present coronavirus disease (COVID-19). Lymphopenia and monocytopenia are important contributors to COVID-19 immunopathogenesis. The fine mechanisms underlying these phenomena are still unknown, and several hypotheses have been raised, some of which assign a role to cell death as far as the reduction of specific types of immune cells is concerned. Thus, we discuss three major pathways such as apoptosis, necroptosis, and pyroptosis, and suggest that all of them likely occur simultaneously in COVID-19 patients. We describe that SARS-CoV-2 can have both a direct and an indirect role in inducing cell death. Indeed, on the one hand, cell death can be caused by the virus entry into cells, on the other, the excessive concentration of cytokines and chemokines, a process that is known as a COVID-19-related cytokine storm, exerts deleterious effects on circulating immune cells. However, the overall knowledge of these mechanisms is still scarce and further studies are needed to delineate new therapeutic strategies.

## 1. Introduction

At the end of 2019, reports about an unknown disease arrived from Wuhan, China. Patients showed symptoms such as fever, malaise, dry cough, and dyspnea, and the diagnosis was viral pneumonia. After a whole-genome sequencing procedure, in January 2020 the virus was identified: a novel coronavirus, to date the seventh member of the coronavirus family capable of infecting humans [1]. Firstly, the World Health Organization named it as 2019-nCoV [1]. Later, as a genetic relation with the coronavirus (CoV) responsible for the Severe Acute Respiratory Syndrome (SARS) epidemic occurring in 2003 was observed, the virus was renamed as SARS-CoV-2 by the International Committee for Taxonomy of Viruses [2]. The origin of this virus was associated with the Huanan South China Seafood Market, where a large variety of animals are sold and where the majority of the first patients worked or passed by [3]. Then, evidence of human to human transmission became strong and the disease rapidly spread around the world. On 11 March 2020, the WHO made a pandemic declaration. From that moment each country faced an unprecedented emergency and established various and different measures to contain the spreading of the virus. As of 13 May 2021, the total number of globally reported cases was over 158 million with more than 3 million deaths (for constantly updated information, see: https://covid19.who.int accessed on 13 May 2021).

## 2. SARS-CoV-2: Structural Characteristics and Diffusion

SARS-CoV-2 belongs to the Coronaviridae family. These viruses have been classified into four genera, namely α-, β-, γ-, δ-Coronaviridae. SARS-CoV, MERS-CoV, and SARS-CoV-2 belong to the β-coronaviruses and all of them are zoonotically transmitted and spread among humans through close contact [4]. SARS-CoV-2 presents a single stranded positive-sense RNA (+ssRNA) of approximately 29.9 kb [5]. It contains four structural proteins (Spike, Envelope, Membrane, Nucleocapsid) and sixteen non-structural proteins (nsp1-16) [6]. SARS-CoV-2 entry into host cells is driven by using the spike glycoprotein (S). This is a transmembrane protein that forms homotrimers that protrude from the viral surface and permit binding to host receptors. After attaching to the host receptor, S goes through a cleavage driven by host proteases that activate the two subunits of the protein, S1, and S2. The first one has the function to bind the host receptor while the latter contains the fusion peptides (FP), essential for the fusion of virus and host membrane that complete the entry process [7]. Once entered into the host cell, SARS-CoV-2 expresses and replicates its genomic RNA to produce full-length copies that will be incorporated into viral particles. The role of the other structural proteins during the process of virus assembling has not been clarified yet [4].

The main receptor identified for SARS-CoV-2 is angiotensin-converting enzyme-2 (ACE2), the same molecule which can bind SARS-CoV. Its identification was supported by the high genomic and structural homology between the two viruses [8,9,10]. ACE2 is a homologue of the angiotensin-converting enzyme involved in the renin-angiotensin-aldosterone system (RAAS), and it regulates blood pressure and fluid balance. ACE2 plays an important role at the lung level, where, for example, it inhibits vasodilation and elevation of vascular permeability. Another important role of ACE2 is at the gastrointestinal level, where it mediates different functions like the regulation of the local innate immunity or the gut microbial ecology. ACE2 is widely expressed in organs and scarcely present in circulation. This molecule can be found in numerous tissues and organs such as the lung, kidney, heart, gut, brain, and others [11]. The expression levels of ACE2 change during the life of each individual, correlating not only with age but also with other diseases such as hypertension, diabetes, or cardiovascular diseases. A correlation between the expression of ACE2 and the severity of the disease caused by SARS-CoV-2 is currently under investigation [11].

Concerning SARS-CoV-2 transmission, the primary mechanism is via fomites and droplets during close and unprotected direct contact between infected and uninfected individuals. The virus can also spread through indirect contact transmission, for example by touching the mouth, nose, or eyes after an indirect exposure [4]. The possibility of transmission through aerosol is still under investigation. The main strengths of this virus are the period of incubation and the possibility to infect through asymptomatic patients. The incubation period is defined as days elapsed between contact and illness-onset. The analysis of cases reported from different countries still revealed difficulties determining the exact period of incubation. Nevertheless, it has been estimated that after an event of exposure, symptoms can occur after 5.1 days (median value) and nearly all infected persons have symptoms within 12 days [12]. This phenomenon leaves the possibility for individuals to spread the virus unaware of being infected. Furthermore, not all infected patients show symptoms: the infection can result in mild to severe symptomatic disease but it can also be asymptomatic [12]. These two factors are crucial for the virus survival, as people that do not know to be infected or in the incubation phase can infect other people.

The most common symptoms associated with viral pneumonia caused by SARS-CoV-2 are fever, cough, sore throat, headache, myalgia, fatigue, and dyspnea. Furthermore, loss of taste or smell and gastrointestinal symptoms are reported in patients [13,14,15]. Disease severity seems associated with various factors such as age, sex, but mostly overall health condition; people with co-morbidities such as diabetes, cardiovascular problems, or hypertension are more likely to experience more severe effects of SARS-Cov-2 infection [16,17].

## 3. Activation of the Immune Response Driven by SARS-CoV-2

The first line of defense of our immune system after contact with pathogens are the physical barriers like the skin and the mucosa lining in the respiratory tract. Epithelial cells infected by the virus produce interferon (IFN), activating the transcription of interferon stimulated genes (ISGs), thus allowing a robust innate immune response. The main actors of the first response are dendritic cells, macrophages, monocytes, and neutrophils, and the intensity of the response to SARS-CoV-2 depends on their activation [18]. Recent studies have suggested that COVID-19 causes alterations in both the lymphocyte and myeloid compartment, with a preferential loss in CD8+ T cells (but also in CD4+), increased plasmablasts, neutrophil expansion, decreased plasmacytoid dendritic cells (pDCs), and differential T cells activation [19,20,21,22]. These cells can migrate to the site of the infection to fulfill their action fighting the pathogen [18].

The vast majority of inflammatory cells infiltrating the lungs are monocytes and macrophages. Lungs autopsy reported the presence of monocytes, macrophages, and a moderate amount of multinucleated giant cells associated with a diffuse alveolar injury [18]. Several molecules involved in monocytes regulation and migration, including tumor necrosis factor (TNF), interleukin (IL)-6, and chemokine CC motif ligand-2 (CCL2), are over-expressed in plasma of COVID-19 patients rather than in age-matched healthy patients, thus revealing an abnormal inflammatory response. Furthermore, high levels of granulocyte-macrophage colony-stimulating-factor (GM-CSF) indicated the presence of emergency myelopoiesis, a process characterized by an increase in myeloid cell differentiation and a consequent mobilization of immature myeloid cells in peripheral blood in response to danger signals [23,24,25]. Following this first activation, a massive release of cytokines produced by various cell types, named "cytokine storm", takes place, that involves several molecules, not only including pro- and anti-inflammatory cytokines of T helper 1 or T helper 2 (TH1 or TH2) type and growth factors but also cell death factors. This indicates the presence of a massive immune activation that indiscriminately involves all cells, similarly to what happens during sepsis [26]. Furthermore, even the concentration of galectins, a family of molecules with different functions, markedly changes. In COVID-19 patients, galectin (Gal)-1, -3, and -9 were increased compared to healthy controls [26]. Gal-1 is a repressor of a number of innate and adaptive immune programs. Gal-3 and -9 act as alarmins, amplifying the immune response in various infectious diseases, including sepsis.

In addition, an important increase of IFN-*γ* was observed in plasma from COVID-19 patients. This molecule activates macrophages which then produce pro-inflammatory cytokines in a process that overwhelms the immune system. Moreover, circulating monocytes showed an important expression of immune checkpoints proteins (ICPs) with the characteristics of being inhibitory such as programmed death-1 (PD-1) and its ligand PD-L1 [23]. The expression and interaction of these molecules, which occurs in cases of exaggerated activation, blocks the effector action of T cells leading to the inhibition of immune response [27].

It is debated whether the action of adaptive immune response could be more helpful or pathogenic in a context of severe COVID-19 in elderly patients. A low frequency of naïve T cells, typical of the aging process, is an element of risk associated with severe COVID-19 [28]. Furthermore, in elderly patients, the immune evasion performed by SARS-CoV-2 could be exacerbated by reduced function or availability of APCs. In such cases, it could be believed that a late T cell response may amplify pathogenic inflammatory outcomes more than overcoming it when a high viral load in the lungs is present [29]. A study on B cells of patients with ages ranging from 37 to 81 years, showed an important alteration of this compartment caused by SARS-CoV-2 infection [30]. Memory B cells, which are supposed to remain stable or slightly increase with aging, were significantly lower in COVID-19 positive patients compared to healthy and younger controls [30]. An increased percentage of transitional B cells was observed, migrating from bone marrow to peripheral lymphatic organs where they maturate into mature-naïve B cells. Furthermore, an increase of plasmablasts was reported, suggesting the involvement of immature B cells in antibody production [30]. All these data underline an immune system’s effort to fight the infection and make up for lymphopenia.

Severe COVID-19 patients, besides the age, show a reduction in CD4+ and CD8+ lymphocytes absolute number, while T cells subsets and CD4+/CD8+ ratio remain stable [31]. The counts of total T cells (CD4+ and CD8+) were significantly lower in Intensive Care Unit (ICU) patients than in non-ICU cases. In addition, statistical analysis showed that severe and critical as well as perished patients had lower T cell numbers than mild/moderate ones [31]. Therefore, the profound T cell loss occurring during SARS-CoV-2 infection seems to correlate with the severity of the disease, and recovery of lymphocytes count has been described as one of the first signs to appear in severe and critical patients just before their health improvement and discharge [31].

Lymphopenia was an important mark also for SARS and MERS, and for both those pathologies was correlated with disease severity. Mechanisms responsible for this condition in patients affected with COVID-19 are still unclear. Glucocorticoid treatment causes lymphopenia as it induces migration of lymphocytes from peripheral blood primarily to lymphoid organs, including bone marrow [32]. At the same time, the viral infection results in the upregulation of endogenous corticosteroids, which may be involved in COVID-19 lymphopenia. Other than a possible migration of lymphocytes, it can be hypothesized as a direct action of SARS-CoV-2, that could infect these cells. This phenomenon would lead to a process of lymphocyte cell death being responsible for lymphopenia [33].

Together with the lymphoid compartment, also the myeloid one is severely affected by SARS-CoV-2 infection. The activation profile and the subset distribution of myeloid cells in COVID-19 patients have been reported to vary at different stages of the disease. In particular, both COVID-19 patients that do or do not require mechanical ventilation and consequently admission to ICU showed a decrease in the absolute number of plasmacytoid and myeloid dendritic cells as compared to healthy subjects [34]. A significant reduction of non-classical monocytes (CD14^low^CD16^high^) was also found in peripheral blood from severely infected patients compared to the mild or control groups, together with a downregulation of HLA-DR on CD14^high^ circulating monocytes [25]. Since IL-6 inhibits HLA-DR expression, an overproduction of IL-6 can cause this phenomenon in CD14+ monocytes of SARS-CoV-2 patients. In the severe disease, the restoration of normal levels of this molecule and the reappearance of the non-classical monocyte subset in the blood was observed in response to anti-IL-6R antibodies [35]. The decrease of non-classical monocytes is not observed in other viral infections, thus generating a highly characteristic feature of COVID-19, also correlated to the severity of infection [36]. The presence of a reduced proportion of non-classical monocytes in peripheral blood could be partially explained by their recruitment to the lungs [37].

Among studies that have highlighted abnormalities in monocytes during SARS-CoV-2 infection, monocyte morphological anomalies in severe COVID-19 patients also affected by type 2 diabetes (T2D) were investigated. A significant loss of classical monocytes, likely due to increased apoptosis, was described [38]. Monocytopenia in T2D COVID-19 patients was accompanied by an exaggerated switch from classical to intermediate and non-classical phenotypes, typical of chronic inflammation and viral infection. The increased expression of CD16 on switched monocytes compared to quiescent cells, which contributes to the induction of CD4+ T-cell polarization, may partially explain the hyperinflammatory syndrome that occurs in diabetic patients characterized by a significant loss of quiescent monocytes [38].

## 4. Cell Death during Viral Infections

All multicellular organisms make a constant effort to maintain homeostasis through the regulation of cellular proliferation, differentiation, and cell death. The correct activation or inhibition of these mechanisms is determinant for the control of cell number and tissue size and for the protection against potential injuries such as those deriving from viral infection. Cell death is the process that allows the removal of damaged or infected cells to create a better environment for healthy cells to perform their functions. Three main mechanisms of programmed cell removal also involved in viral response are apoptosis, pyroptosis, and necroptosis. These are energy-dependent mechanisms that involve a variety of molecular pathways and regulatory genes [39].

### 4.1. Apoptosis

Apoptosis plays a central part in normal tissue homeostasis and is highly regulated by several molecular pathways whose correct or incorrect activation determines physiological or pathological outcomes [40]. Two are the main pathways that control the initiation of apoptosis: the extrinsic or death receptor pathway and the intrinsic or mitochondrial pathway [41]. The extrinsic or death receptor signaling pathway involves death receptors, that belong to the tumor necrosis factor (TNF) receptor gene superfamily [42]. Two of the best characterized members are TNF-R1 and Fas (CD95). The interaction of FasL (CD95L) with Fas coordinates the clustering of the receptors and the recruitment of adaptor protein FADD and pro-caspase 8 to assemble the death-inducing signaling complex (DISC) [43]. Active caspase 8 can directly cleave caspases 3/7 or process the BH3-only member BID, releasing a truncated form of BID, which translocates to the mitochondria and induces mitochondrial outer membrane permeabilization (MOMP), with consequent release of apoptogenic factors such as cytochrome c (cyt c) and the assembly of the apoptotic protease activating factor-1 (Apaf-1)/caspase 9 apoptosome, responsible for the activation of executioner caspases 3, 6 and 7 [43] (Figure 1). The intrinsic or mitochondrial pathway is triggered by stimuli such as DNA damage that trigger permeabilization of mitochondria and release of cyt c into the cytoplasm [44]. This converges on the formation of the apoptosome and subsequent caspases activation (Figure 1) [41].

The dysregulation of apoptosis is a feature of a wide variety of diseases. Uncontrolled apoptosis is found in infertility, immunodeficiency, and chronic degenerative diseases such as human neurodegenerative diseases, Parkinson’s and Alzheimer’s, whereas delayed or inhibited apoptosis is a hallmark of cancer and autoimmunity [45]. Apoptosis occurs also during viral infections and normally leads to a beneficial outcome for the infected host, but deleterious for the invading virus whose replication is blocked in the absence of a viable susceptible host cell. However, it is also possible that virally induced apoptosis proves harmful to the host, damaging immune cells that should protect it. For example, as demonstrated by Baloul and colleagues, the increased FasL production after rabies infection in mice induces migratory T cell death via the extrinsic apoptotic pathway, making the host unable to control the viral invasion in the central nervous system. Whereas, in mice lacking FasL the phenotype of the infection was less severe with a reduced number of CD3+ T cells undergoing apoptosis compared to normal mice [46]. In addition, viruses can take advantage of apoptosis as during lytic infection, or at late stages of infection, this type of cell death can permit the dissemination of their progeny [47]. Overall, through apoptosis, viruses can create an environment of death and inflammation, and apoptosis can be considered a double-edged sword. Viruses usually upregulate cell death receptors or their ligands on the cell surface and increase the sensitivity of cells to death receptor-mediated apoptosis. Even if the principles of the mechanisms are quite common to different viruses, each of them acts in a specific way [47]. Regarding Human Immunodeficiency Virus (HIV), increased Fas antigen expression on CD4+ and CD8+ T lymphocytes has been observed and linked to their depletion [48,49,50]. Moreover, HIV-1 infection induces the expression of TRAIL and TRAILR2, leading to TRAIL-mediated apoptosis in CD4+ T cells. This process is driven by HIV-1 stimulated pDCs, that in response to the infection produce IFN-α which regulates the expression of TRAIL [51,52]. TRAIL induces apoptosis also in memory B cells [53]. In chronic hepatitis C virus (HCV) infection, Fas expression increases on peripheral blood mononuclear cells (PBMCs), and human hepatocytes are sensitized to TRAIL-induced apoptosis [54]. This mechanism seems to correlate with liver pathology and fibrogenesis [55,56]. The same effects were observed during the infection with hepatitis B (HBV) [57]. In the family of Herperviruses, varicella-zoster virus (VZV) activates caspase-8 in melanoma cells [58]. Influenza virus infection induces co-expression of Fas and FasL on the surface of infected cells, so apoptosis can occur when cells come in contact with each other [59]. MERS-CoV is known to induce apoptosis in different types of cells. This virus prompts alveolar cell death together with extensive infection of the huge alveolar surface; this is consistent with acute lung injury observed in MERS-CoV-infected humans [60]. MERS-CoV also infects T cells, inducing the activation of the intrinsic and extrinsic apoptosis pathways. In fact, the case fatality rate of this virus is higher than both SARS and COVID-19 [61].

Viruses not only promote host cell apoptosis to release viral particles but in some cases inhibit the process to totally exploit cells for their replication. So even if host cells promote apoptosis to stop virus replication, viruses have elaborated different mechanisms to inhibit this process, for example by expressing death receptors orthologs or encoding several proteins that antagonize host cell death signals [47]. An example is given by human cytomegalovirus (hCMV) infection, in which latently infected myeloid progenitors become refractory to Fas-mediated killing allowing the virus to survive and maintain its reservoir [62,63].

### 4.2. Necroptosis

Necroptosis is a pivotal mechanism of programmed cell death used to contrast environmental stresses that can be caused by chemical and mechanical injury, inflammation, and pathogen-mediated infections. If apoptosis is a pathway that does not fuel inflammation and could be defined as a “contained” and immunologically silent form of cell death, necroptosis follows a completely different strategy [64], exhibiting features of both apoptosis and necrosis. However, it is inhibited by necrostatin-1 (Nec-1), a molecule that blocks the activity of receptor-interacting protein kinase (RIPK) 1 [65]. This suggests that necroptosis is a well-regulated process, and, for this reason, is also defined as a programmed necrosis [66]. Necroptosis can be considered a form of regulated necrosis mediated by death receptors. The relation between necroptosis and inflammation has been observed in different diseases and is the main component of the pathogenesis of necroptosis-associated diseases [67,68]. Upon TNF binding to its cognate receptor, TNF receptor type 1-associated death domain protein (TRADD), TNF receptor-associated factor 2 (TRAF2), Fas-associated protein with death domain (FADD), RIPK1, and caspase-8 are recruited. If caspase-8 is compromised, RIPK1 interacts with RIPK3 to trigger consecutive downstream signaling events as the recruitment and phosphorylation of mixed lineage kinase domain-like (MLKL). Then, pMLKL oligomerizes and translocates towards the plasma membrane from cytosol resulting in the formation of the pore [66,69] (Figure 1). Therefore, two are the basic conditions for necroptosis to happen: (i) cells must express RIPK3, and (ii) caspase-8 must be inhibited. Thus, necroptosis takes place when the apoptosis pathway faces impediments [66].

Necroptosis is a crucial host defense mechanism to fight pathogens. Indeed, it could be used both for viral clearance in infected cells and to create an inflammatory environment with the release of damage associated molecular patterns (DAMPs), to recruit immune system cells and cytokines or chemokines to fight the infection. In response, viruses have developed different strategies to neutralize host programmed cell death, mostly known to block the apoptosis pathway. In those cases, the activation of the necroptosis pathway can still rescue the host and fight the infection. However, little is known about the interaction between necroptosis prompted by the host and virus escaping mechanisms [64]. For example, large DNA viruses inhibit necroptosis through the inactivation of caspases, promoting viral persistence and immune evasion. Conversely, Poxviruses encode serpins which inhibit caspase-1 and -8. This can block apoptosis, but it provides a natural priming signal for inducing necroptosis [70]. hCMV suppresses necroptosis induced by TNF and caspase inhibitors by interfering in the pathway after MLKL pseudokinase activation [71]. Epstein–Barr Virus (EBV), a γ−herpesvirus, seems to interact with RIPK1 and RIPK3 through its latent membrane protein 1 (LMP1), modulating the ubiquitination status of those proteins [72]. Regarding HIV-1, the induction of necroptosis was characterized by pMLKL expression in CD4+ T cells driven by HIV-1 strain with C-X-C Motif Chemokine Receptor 4 (CXCR4) tropism, demonstrating that HIV can induce necroptosis in CD4+ T cells [73]. The inhibition of necroptosis, then, is crucial for virus propagation; mechanisms that could be responsible for this process are not well known yet and more investigation is needed [65].

cFLICE-like inhibitory protein (FLIP) is also involved in necroptosis. There are two forms of this protein: cFLIPL and cFLIPS. The former harbors mutations at the C-terminal caspase domain, losing its enzymatic activity, thus promoting caspase-8 activation through heterodimerization. The latter lacks the C-terminal caspase domain, thus inhibiting caspase-8 activity [74]. However, a more complicated scenario is present in certain viral infections. Indeed, a class of viral FLIP proteins (vFLIP) has been identified in herpesviruses and in the Molluscum contagiosum virus (MCV) [75,76]. vFLIP lacks the C-terminal caspase domain, resembling the human cFLIPS [77,78]. Surprisingly, vFLIP also exhibits an inhibitory activity, interacting with RIPK1 [79]. In Kaposi’s sarcoma-associated herpesvirus (KSHV) the role of vFLIP has been accurately investigated, and it was found that it induces nuclear factor kappa B (NF-kB) signaling thus inhibiting apoptosis by inducing the expression of cellular pro-survival factors. The suppression of vFLIP induces apoptosis of infected cells, demonstrating that this protein in KSHV is fundamental for the survival of infected lymphoma cells [80]. This viral protein, which is structurally similar to the human cFLIPL, inhibits the programmed cell death pathways for longer survival of infected cells; this is another viral mechanism used to exploit cells for viral replication and survival.

### 4.3. Pyroptosis

Pyroptosis is a form of cell death dependent on caspase activity which differs from apoptosis for its inflammatory outcome. Pyroptosis was initially identified in macrophages infected with *Salmonella typhimurium* and *Shigella flexneri* and has been characterized as a pro-inflammatory cell death marked by rapid lysis of the cell, with the release of the intracellular content including immunostimulatory molecules such as IL-1β and IL-18 [81,82,83]. The inflammatory outcome depends on the activity of caspase-1 which is responsible for the activation and release of these inflammatory mediators.

Caspase-1 cleaves the precursors of IL-1β and IL-18 into their mature active forms and gasdermin (GSDM)-D, a member of the pore-forming gasdermin gene family. GSDM-D cleavage is necessary for the formation of the membrane pore, which is essential for the secretion of mature IL-1β and IL-18. Caspase-1 activation is triggered by different extracellular or intracellular signals such as invading pathogens (PAMPs), DAMPs, or toxins, whose recognition by specific Nod-like receptors (NRLs) results in the formation of inflammasomes. The NLR family pyrin domain-containing 3 (NLRP3) inflammasome is the best-studied inflammasome involved in pyroptosis in which caspase-1 is recruited and activated (Figure 1). The subsequent production of inflammatory mediators stimulates the recruitment of immune cells and the development of adaptive immune response that leads to the resolution of infection. IL-18 plays an important role in the differentiation of CD4+ T cells in their T-helper 1 phenotype (TH1) and activation, as demonstrated in infected mice deficient for caspase-1, that generated an impaired TH1 response compared to wild type counterpart [84].

Several viruses can trigger pyroptosis through the activation of inflammasomes. NLRP3 inflammasome is activated by different families of viruses, including RNA and DNA viruses. In mice infected with RNA viruses, such as Sendai virus and Influenza A virus, viral dsRNA activates caspase-1 in macrophages. In addition, IL-1β production in murine dendritic cells and macrophages after infection with encephalomyocarditis virus (EMCV) or vesicular stomatitis virus (VSV) is dependent on NLRP3 inflammasome [85,86]. The infection with West Nile virus (WNV) leads to increased production of IL-1β which is required for the migration of Langerhans cells to lymph nodes [87]. Also, IL-1β signaling and the NLRP3 inflammasome activation were identified as key host restriction factors for the control of WNV infection in the central nervous system [88]. NLRP3 activation and caspase-1-dependent pyroptosis occurs also after infection with the HCV [89].

Regarding HIV infection, researchers questioned the involvement of pyroptosis in the CD4 T-cell depletion since caspase-1 was found activated in lymphoid CD4+ T cells after HIV infection [90]. The treatment of infected cells with various inhibitors revealed that blocking caspase-1, but not caspase-3, efficiently prevented the depletion of CD4+ T cells. These inhibitors also blocked the release of the cytosolic enzyme lactate dehydrogenase that occurs after the membrane rupture during pyroptosis. Moreover, a large production of IL-1β was detected in lymphoid CD4+ T cells infected with HIV. The ex vivo staining for caspase-1 of lymph node tissues from a patient positive for R5-tropic HIV revealed a high enzyme activity in the surrounding paracortical zone, which typically hosts resting CD4+ T cells. Here, high amounts of IL-1β were also detected. The staining to detect the active form caspase-3 was limited to the areas of permissive and activated CD4+ T cells in germinal centers. Therefore, it was confirmed that an intense pyroptotic death occurs in non-permissive quiescent CD4+ T cells from infected lymphoid tissue. It is important to consider the role of pyroptosis in chronic inflammatory states related to HIV infection. The continuous release of inflammatory molecules from dying lymphocytes attracts more cells to the infected lymphoid tissues, which in turn die and release more molecules. This leads to chronic inflammation, that together with CD4+ T depletion, promotes disease progression [90].

Regarding DNA viruses, studies reported the activation of NLRP3-dependent inflammasome and downstream IL-1β production after the internalization of adenovirus DNA in murine macrophages. Similarly, infection with herpes simplex virus 1 (HSV-1), a DNA virus of the Herpesviridae family, caused strong caspase-1 activation and pro-IL-1β maturation [91]. In human monocyte-derived macrophages, activation of the NLRP3 inflammasome is dependent upon Adenovirus type 5 (Ad5) penetration of endosomal membranes and the release of the lysosomal protease cathepsin B into the cytoplasm. Ad5 cell entry also induces the production of reactive oxygen species (ROS), which are required for NLRP3 activation [92]. Additionally, VZV is sensed by NLR and activates NLRP3 inflammasome both in vitro, in VZV-infected THP-1 (a transformed monocyte cell line), primary lung fibroblasts, and melanoma cells, and in vivo, in human VZV-infected skin xenografts in SCID mice [93].

Active caspase-1 allows the host to control several microbial infections. Therefore, it is not surprising that pathogens have evolved mechanisms to limit the activation of caspase-1 in response to infection. An example is given by Poxviruses, DNA viruses that replicate in the cytoplasm. To avoid their detection by NLRP3, M13L-PyD binds apoptosis-associated speck-like protein containing a CARD (ASC), determining the disruption of inflammasome and preventing the recruitment and activation of caspase-1 [94]. The influenza virus protein NS1 can also limit caspase-1 activation and cell death [95]. On the one hand, these data suggest that the inhibition of caspase-1 is a strategy widely adopted by pathogens to persist in infected cells. On the other hand, host cells have evolved defensive mechanisms to control infections, which aim to limit pathogen replication, enhance immune responses and improve host survival [96]. Although these pathways contribute to pathogen clearance, an inappropriate caspase-1 activation, due to mutations in NLR proteins or to the presence of inflammatory stimuli, can be deleterious. Caspase-1 is involved in many diseases, including myocardial infarction, cerebral hypoxia, endotoxic shock, and neurodegenerative diseases. Pharmacological inhibitors of caspase-1 protect against excessive inflammation and the resulting tissue damage and dysfunction [96].

## 5. Cell Death during SARS-CoV-2 Infection

SARS-CoV-2 infection results in hyperactivation of the immune system, leading to a cytokine storm [26]. This phenomenon is directly correlated with lung injury, multiple organ failure, and critical prognosis [97,98]. The mechanism responsible for this outcome is still to be clarified but cell death pathways seem to fairly link the two aspects of the disease. Among those, the role of apoptosis, pyroptosis, and necroptosis during SARS-CoV-2 infection will be here deepened.

### 5.1. Apoptosis

The role of cell death and apoptosis in COVID-19 is a topic still under investigation. Intrinsic and extrinsic pathways are induced by SARS-CoV-2 infection [99]. In this field, studies on SARS-CoV-2 have mirrored those on SARS-CoV. The role of Open Reading Frame 3a (ORF3a), a conserved coronaviruses accessory protein, was investigated [100,101]. ORF3a is a viroporin, a transmembrane protein acting as an ion channel, whose function has been associated with the capacity to contribute to virus release [102,103,104]. In fact, in animal models, the deletion of this protein reduced viral replication [105]. Experiments on various cell lines revealed that ORF3a induces the cleavage and activation of caspase-8 without affecting Bcl-2 [101]. This means that ORF3a induces apoptosis via the extrinsic pathway. Inhibitors of caspase-8 or caspase-9 significantly suppress ORF3a-induced apoptosis. ORF3a is also present in SARS-CoV-2 and shares 73% homology with its SARS-CoV counterpart [101]. Mutations that lead to a cytosolic form of the protein eliminates the capacity of inducing apoptosis only for SARS-CoV-2, whereas the membrane-associated feature is involved but not essential in triggering this pathway for SARS-CoV [101]. This likely suggests that SARS-CoV-2 has a weaker pro-apoptotic activity if compared to SARS-CoV and that this difference can partially explain the different pathogenicity of these viruses. Indeed, SARS-CoV-2 has been generally believed to be less virulent than SARS-CoV and to lead to the generation of asymptomatic subjects which helps in a major spreading of the virus [101].

An important reduction of CD4+ and CD8+ T-cell subsets that do not significantly impact the CD4+/CD8+ ratio but on the absolute numbers is an important feature of SARS-CoV-2 infection [106]. Lymphopenia is a risk factor in viral infections and has been associated with the severity of the disease in COVID-19 [106]. The mechanisms underlying lymphopenia are still unknown, but different hypotheses have been raised. According to one of these, the cytokine storm can lead to a pro-inflammatory status which, in turn, induces lymphocytes apoptosis [106]. There is also evidence of the involvement of CD95 in the induction of apoptosis in lymphoid cells of SARS-CoV-2 patients. CD95 expression increased in circulating CD4+ and CD8+ lymphocytes of infected subjects compared to healthy controls, and a direct correlation between higher CD95 expression and lower CD4+ absolute count was observed. A similar pattern was detected among CD8+ T cells [107]. A recent study reported that after RNA-sequencing of PBMCs from COVID-19 patients several genes were altered, including those related to apoptosis and P53 signaling; those genes were highly expressed in patients with COVID-19 compared to healthy donors [108]. In conclusion, apoptosis could be responsible, at least in part, for the reported lymphopenia of COVID-19 infected patients.

Single-cell RNA sequencing (scRNA-seq) of CD3+ T cells from acutely ill COVID-19 patients revealed an upregulation of cell death programs genes [109]. These data also evidenced CD3+ T cells mitochondrial dysfunction, including the downregulation of several programs associated with mitochondria organization and function. Electron microscopy revealed that mitochondria of lymphocytes from COVID-19 patients were dysmorphic, had irregular shapes, and incomplete *cristae* [109]. Cyt-c was found in the cytoplasm of lymphocytes from COVID-19 patients, confirming the presence of damaged mitochondria. This intracellular condition can be the sign for the activation of the intrinsic apoptotic pathway that could partially contribute to the lymphopenia developed by COVID-19 infected patients [109].

Regarding MERS-CoV, T cells were highly susceptible to infection and consequently apoptosis. Moreover, this infection caused the dramatic depletion of the surface receptor dipeptidyl peptidase 4 (DPP4) impacting T cells proliferation and functionality and so inducing apoptosis [110]. The role of CD147 as another possible entry receptor for SARS-CoV-2 was investigated, taking into account the possibility that lymphopenia due to T cell apoptosis could be caused by virus entry [111]. CD147, also termed as EMMPRIN or basigin, is a transmembrane glycoprotein of the immunoglobulin superfamily with high pleiotropism [112]. CD147 is a receptor for several viruses: it can mediate the entry of HIV into T lymphocytes, the entry of CMV into endothelial and epithelial cells, and the entry of Plasmodium falciparum into red blood cells. The entry of SARS-CoV-2 into lymphocytes would cause an excessive secretion of pro-inflammatory molecules that indirectly upregulates P53 apoptotic pathways, so resulting in cell death [111]. In vitro inhibition of CD147 with meplazumab significantly blocked virus replication [113], and overexpression of this receptor in cell lines promoted virus infection. SARS-CoV-2 can enter the host cells through CD147-mediated endocytosis rather than membrane fusion [113]. In conclusion, the role of CD147 needs more investigations to better understand whether and how SARS-CoV-2 uses this molecule to enter into lymphocytes and directly promote apoptosis.

Consistent with different studies, pDCs proportion in blood was lower in COVID-19 patients compared to healthy control [20,21,114]. However, there is no evidence of pDCs migration into tissues after SARS-CoV-2 infection, neither infiltration was found in bronchoscopy [115,116,117,118]. pDCs are one of the major producers of type I IFN and orchestrator of T- and B-cell responses upon viral infection [119]. As previously reported, in severe COVID-19 infections, a decrease in type I IFN signature was observed [120]. To better investigate the relationship between IFN-I signature and pDCs, the signature of these cells was studied. The analysis revealed that an apoptotic gene signature, which includes BRCA2, CASP3, CASP8, BID, BAK1, and XBP1, was positively associated with disease status and severity [120]. In addition, the apoptotic gene signature score was negatively correlated with pDC frequency. At the same time, pDC frequency was positively associated with IFN-I signature in different cell populations, indicating that in a situation of decreased pDC frequency, IFN-I signature is directly affected. Therefore, all these findings indicate that apoptosis is responsible for the decreased frequency of pDCs in the blood of COVID-19 patients and that this phenomenon could be the direct cause of the observed decreased IFN-I signature [120].

### 5.2. Necroptosis

Whether SARS-CoV-2 can trigger necroptosis was studied on Calu-3, a human lung cancer cell line. MLKL is the effector of necroptosis and is phosphorylated by RIPK3 [121]. After the infection of Calu-3 with SARS-CoV-2, the phosphorylation of MLKL was studied through Western Blot and immunofluorescence. pMLKL was upregulated in infected cells and also the staining pattern revealed the presence of this protein on the plasma membrane. To confirm that the mechanism was induced by SARS-CoV-2, the infected cells were inactivated using ultraviolet rays. This process prevented virus replication and phosphorylation of MLKL, indicating the viral-dependent activation of the necroptotic pathway. In addition, the inhibition of RIPK3 in infected cells inhibited also pMLKL, indicating that SARS-CoV-2 acts directly on RIPK3 to activate necroptosis [121]. In this scenario, the use of Nec-1/Nec-1 analogues might alleviate cytokine storm, systemic inflammation, and COVID-19 thanks to its activity to block RIPK1, the molecule that interacts with RIPK3 and promotes the phosphorylation of MLKL [122]. As RNA viruses can promote the inflammasome activation via RIPK1/RIPK3 [123], Nec-1 may help in regulating this process by limiting inflammation and cytokines release induced by COVID-19.

Whether the inhibition of necroptosis favors host antiviral response or aggravates tissue inflammation and damages is not yet validated. In fact, necroptosis can both block viral replication by inducing cell death but also promote viral spreading by cell rupture [122]. Moreover, during Influenza A, the activation of the necroptotic pathway did not depend on RIPK1 [124]. Thus, using Nec-1 to control cytokine storms does not necessarily compromise host antiviral response [122]. Finally, it was observed in COVID-19 severe patients that T lymphocytes become functionally exhausted [26]; thus Nec-1 may also ameliorate this condition by regulating host defense [122].

Necroptosis has been implicated in ARDS development following sepsis or trauma [125], but the relationship between necroptosis and COVID-19-induced ARDS remains unclear. The plasma level of RIPK-3 was measured in a cohort of severe and mild COVID-19 patients. It was observed that the median level of serum RIPK3 was higher in severe patients compared to mild ones. These results suggested that RIPK3 signaling pathways are involved in the worsening of the pathology and maybe also in the development of acute lung injury associated with COVID-19 pneumonia [126]. Due to the involvement of RIPK3 in other inflammatory pathways [127], a direct implication of SARS-CoV-2 in inducing necroptosis cannot be stated.

### 5.3. Pyroptosis

The role of inflammasome on the activation of pyroptosis is well known. SARS-CoV expresses at least three proteins that could activate NLRP3: envelope (E), ORF8b, and ORF3a. As NLRP3 is sensitive to cytosolic ionic concentration, the virus seems to act through its E-protein viroporin, facilitating Ca2+ leakage into the cytosol, and an ORF3a-mediated extracellular K+ efflux [128,129]. The loss of ionic balance promotes mitochondrial damage and the generation of ROS, which coactivates NLRP3 [128] (Figure 2).

E protein and ORF3a can activate NLRP3 through at least other two mechanisms: (i) by stimulating NF-kB signaling to activate the transcription of different inflammatory chemokines and cytokines; (ii) by promoting TRAF3-mediated ubiquitination of ASC [130,131,132,133]. It is still unknown if SARS-CoV-2 is able to activate NLRP3. However, considering that ORF3a and E share 73% and 94.7% homology to SARS-CoV, respectively, it can be hypothesized that the same, or at least similar mechanisms, occurs during COVID-19 [134]. NLRP3 inflammasome, a powerful pro-inflammatory platform, is expressed in various cell types as innate immunity, endothelial, hematopoietic, lung epithelial, kidney, and cardiac cells [135]. Evidence reported that NLRP3 activation in these cells is a response to angiotensin II stimulation binding to angiotensin II receptor type 1 (AT1) [136,137,138].

ACE2, the main receptor for SARS-CoV-2, has been found on the surface of different cell types. Its role is to convert angiotensin II (Ang II) to Ang (1–7) [139,140,141]. These proteins are a member of the RAAS system, but their effects are opposite. The former binds the angiotensin 1 receptor (AT1R) while the latter activate Mas Receptor (MasR) [142]. The activation of AT1R after SARS-CoV-2 infection has detrimental outcomes such as the induction of fibrosis or the increase in reactive oxygen species (ROS) release. Its hyperactivation, in addition, may lead to NLRP3 inflammasome activation and cell death by pyroptosis. Conversely, the activation of MasR is protective having anti-fibrotic, antioxidant, and vasodilatory effects [142]. ACE2, after binding to SARS-CoV-2, is internalized and it consequently does not complete the conversion of angiotensin II to Ang (1–7), leading to the stimulation of AT1R and the activation of NLRP3 [143]. Among all the types of cells that express ACE2, also hematopoietic cells present this receptor on their surface. It was demonstrated that human Very Small Embryonic-Like Stem Cells (VSELs) express ACE2 indicating that if the aforementioned mechanisms are confirmed, early-development human stem cells residing in adult tissues may be affected by SARS-CoV-2, which could lead to long-term damages [144].

### 5.4. PANoptosis

Cytokine storm is a possible cause of organ damages developed during COVID-19. In particular, TNF and IFN-γ highly upregulated in patients with COVID-19, are critically involved in this process. In a murine model of shock induced by TNF and IFN-γ, a reduction of B and T cells together with a concomitant increase in the neutrophil-to-lymphocyte ratio (NLR) was observed. Interestingly, these alterations are also present in patients with severe COVID-19 [145]. In bone marrow-derived macrophages (BMDMs), TNF and IFN-γ together induced a robust cleavage of gasdermin E (GSDME), a protein that can induce pyroptosis in specific conditions, triggering membrane pore formation [146]. In addition to pyroptosis, the activation of caspase-3, -7, and -9 was also reported, thus indicating that apoptosis was also induced [145].

The activation of the protein kinases RIPK1 and RIPK3 and the subsequent phosphorylation of MLKL have also been observed during TNF and IFN-γ treatment, suggesting that necroptosis can be induced (Figure 2). Thanks to a deeper analysis of the signaling pathway induced by these cytokines, it was concluded that “inflammatory cell death” depended on interferon regulatory factor 1 (IRF1) and nitric oxide (NO) axis. NO induces T cells death, and T cells lacking inducible NO synthase (iNOS) have reduced post-activation death [147,148]. Regarding the IRF1 pathway, T cells lacking STAT1 are resistant to activation-induced pathways, confirming the role of IRF1 in T cell death mechanisms [149]. TNF and IFN-γ, which are present at high concentrations in plasma of COVID-19 patients, may aggravate lymphopenia through direct killing of lymphocytes [145]. Overall, these data suggest that these cytokines together lead to PANoptosis, the phenomenon whereby pyroptosis, apoptosis, and necroptosis are concurrently engaged in the same cell population [150]. The concept of PANoptosis describes a unique inflammatory programmed cell death based on the extensive crosstalk between the main cell death pathways and regulated by the PANoptosome, a single complex that provides a molecular scaffold for the interaction of key components of each pathway. The defined composition of the PANoptosome is not well established and it is likely to depend on the different stimulus that induces its assembly. It can include NLRP3, ASC, caspase-1, which are regulators of pyroptosis, caspase-8 which other than apoptosis, it is also involved in the regulation of the other cell death pathways being recruited by FADD, and kinases RIPK1 and RIPK3, that triggers necroptosis [145,151,152] (Table 1).

## 6. Strategies for Apoptosis Modulation in SARS-CoV-2 Infection

To date, as shown by several studies, apoptosis seems to be the leading cell death mechanism occurring as a consequence of SARS-CoV-2 infection. Accordingly, since apoptosis involves a large number of molecules and processes, various strategies could be used to affect these mechanisms.

In the extrinsic pathway, the activation of caspase-8 prompts the initiation of a caspase cascade that results in cell death [153]. Necessary for its activation is the role of cFLIP in the DISC complex [153]. The long isoform of cFLIP (cFLIPL) at the DISC can act both in a pro-apoptotic and in an anti-apoptotic manner [153,154]. It acts as an anti-apoptotic factor by the inhibition of procaspase-8 activation, therefore viruses upregulate cFLIP expression as a mechanism to block extrinsic apoptosis at the early stage of infection to sustain their replication [154]. High levels of cFLIP were observed at the very beginning of SARS-CoV-2 infection, as well as in the lungs of COVID-19 patients [155]. Therefore, the use of small molecules, that act against cFLIP to activate caspase 8 and induce extrinsic apoptosis, has been proposed as an important strategy to contrast virus replication [156].

Another relevant antiapoptotic pathway that is controlled by viruses, including SARS-CoV-2, is the NF-kB pathway. Activation of NF-kB leads to the upregulation of the most important apoptosis inhibitors, such as cFLIP, Bcl-2, and XIAPs. This strategy would be crucial to promote viral infection and inflammation. According to the analysis of mRNA expression, different components of the NF-kB pathway are upregulated during SARS-CoV-2 infection [157,158]. Hence, selective inhibition of the NF-kB machinery with small molecules, such as proteasome inhibitors, represents another possible solution to target the anti-apoptotic pro-inflammatory response induced by SARS-CoV-2 [159].

In addition, survival of T cells that present features of apoptosis induced by mitochondria degeneration could be rescued with either Voltage Dependent Anion Channel 1 (VDAC1) oligomerization inhibitor or pan-caspase inhibitor [98]. These findings are consistent with the fact that VDAC1 oligomerization and interaction with BCL2 family proteins are thought to be responsible for the formation of pores in the outer membrane of mitochondria, allowing for cyt c release and initiation of caspase cascade, that induces cellular apoptosis [160,161].

Finally, focusing on the cytokine storm, the inhibition of cell death mechanisms by targeting TNF and IFN-γ induced pathway represents the most immediate therapeutic strategy to pursue [145], as already described for drugs able to inhibit IL-6 pathway [162].

## 7. Conclusions

Cell death mechanisms are fundamental for the maintenance of an appropriate environment for healthy cells to perform their functions. Different pathways are involved in this phenomenon.

Among them, apoptosis is a physiological, but also pathological, mechanism of programmed cell death. During viral infections, its role is crucial for the host’s homeostasis. Unfortunately, viruses can regulate and use apoptosis to their advantage. This regulation can take place in a dual way. On the one side, pathogens can promote apoptosis to release viral particles and contribute to general damages to the host. On the other side, as host cells are fundamental for viral replication and particle assembly, pathogens can also inhibit the process of apoptosis to exploit cells. Each virus has developed a specific strategy to control this process. Since the important homology of various structures between β-Coronaviruses, it can be hypothesized that cell death pathways are maintained or are, at least, very similar. The role of ORF3a in inducing the extrinsic pathway similarly to SARS-CoV infection should be deepened. Whether T cells are also affected by SARS-CoV-2 infection should not be excluded.

The possibility to use drugs to control apoptosis may provide future therapeutic opportunities to fight SARS-CoV-2 infection and its downstream pathological consequences. Molecules such as c-FLIP or NF-kB inhibitors or caspase-8 activators could be studied and eventually included among the drugs used to contrast COVID-19. Moreover, the role of mitochondria during apoptosis should be deepened in order to understand and find what is the cause of the organelle dysfunction that leads to the initiation of apoptosis. Also, in this case, the use of mitochondrial protein inhibitors should be considered. In addition, a fast and direct strategy to target the uncontrolled response of the immune system after the infection could be the use of molecules to inhibit the action of TNF and IFN-γ.

In conclusion, SARS-CoV-2 infection triggers several cell death pathways that could become pharmacological targets. Apoptosis, in fact, may have a critical role in the outcome of COVID-19, depending on the cells involved in this type of cell death, and the timing of this phenomenon. Studying cell death during COVID-19 could help to better understand the pathogenesis of this devastasting disease, and to identify promising therapeutic options, eventually able to mitigate the damages caused by a pathogenic immune response.

## Figures and Tables

**Figure 1 cells-10-01585-f001:**
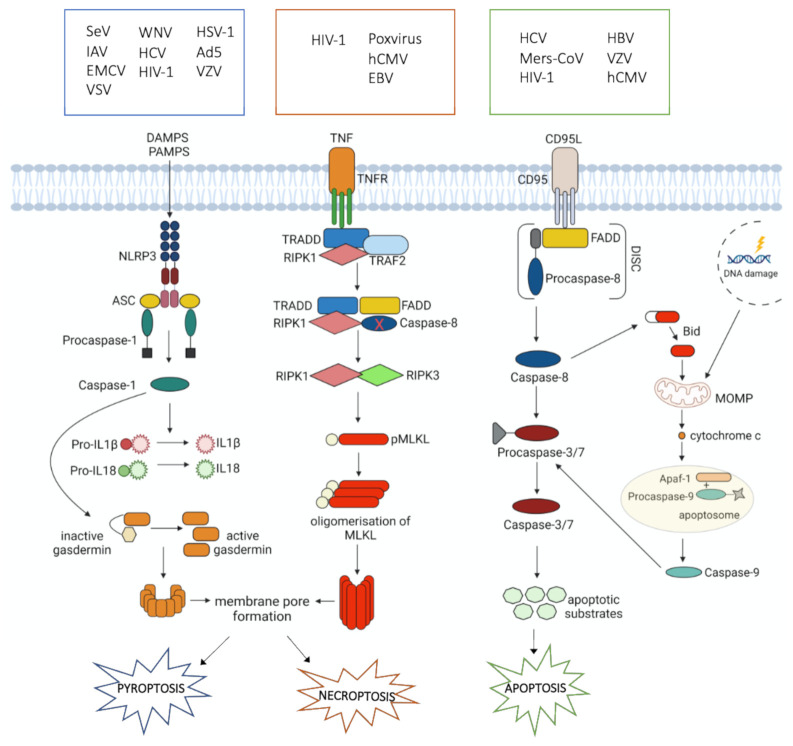
Principal mechanisms involved in pyroptosis, necroptosis, and apoptosis triggered by viral infections. Damage-associated molecular patterns (DAMPS), pathogen associated molecular patterns (PAMPS), NLR family pyrin domain-containing 3 (NLRP3), apoptosis-associated speck-like protein containing a CARD (ASC), interleukin1β (IL1β), interleukin18 (IL18), tumor necrosis factor (TNF), tumor necrosis factor receptor (TNFR), tumor necrosis factor receptor type 1-associated death domain protein (TRADD), Fas-associated protein with death domain (FADD), TNF receptor-associated factor 2 (TRAF2), receptor-interacting serine/threonine-protein kinase 1 (RIPK1), receptor-interacting serine/threonine-protein kinase 3 (RIPK3), p-MLKL: phosphorylated mixed lineage kinase domain-like, CD95 ligand (CD95L), apoptotic protease activating factor-1 (APAF-1), death-inducing signaling complex (DISC), mitochondrial outer membrane permeabilization (MOMP), Hepatitis C virus (HCV), Middle East Respiratory Syndrome Coronavirus (Mers-CoV), Human Immunodeficiency virus 1 (HIV-1), Hepatitis B virus (HBV), Varicella Zoster virus (VZV), Human Cytomegalovirus (hCMV), Epstein–Barr virus (EBV), Sendai virus (SeV), Influenza A virus (IAV), Encephalomyocarditis virus (EMCV), Vesicular Stomatitis virus (VSV), West Nile virus (WNV), Herpes Simplex virus 1 (HSV-1), Adenovirus type 5 (Ad5).

**Figure 2 cells-10-01585-f002:**
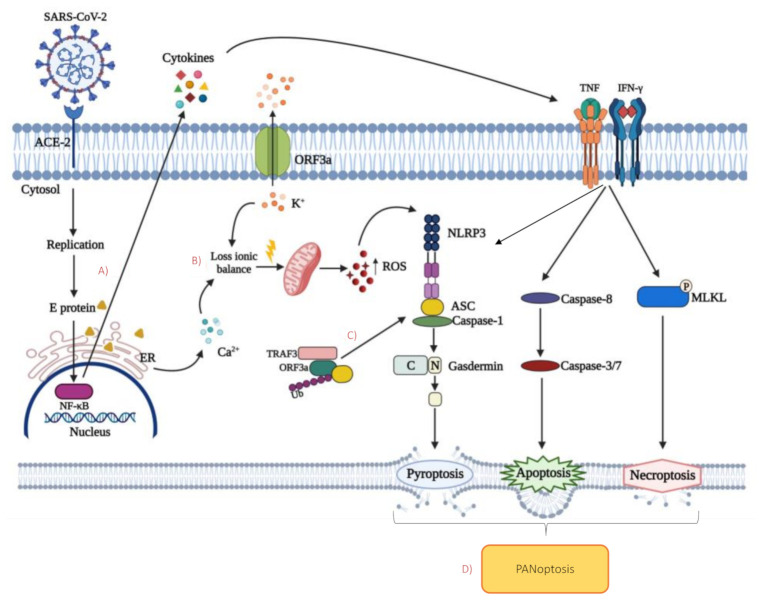
Possible cell death mechanisms induced by SARS-CoV-2 proteins and cytokine production. (**A**) SARS-CoV-2 E protein stimulates the nuclear factor kappa B (NF-kB) signaling pathway, leading to the production and release of inflammatory cytokines. Tumor necrosis factor (TNF) and interferon gamma (IFN-γ), binding their receptor, can trigger three different mechanisms: (1) apoptosis via the extrinsic pathway; (2) necroptosis through the phosphorylation of mixed lineage kinase domain-like (MLKL); (3) pyroptosis by the cleavage of inactive full-length gasdermin into its active form that can form membrane pores. (**B**) E protein contributes to the loss of ionic balance by facilitating Ca2+ leakage from endoplasmic reticulum into the cytosol, together with viral membrane-associated protein ORF3a that mediates the efflux of K+. The consequent mitochondrial damages and reactive oxygen species (ROS) production activate NLR family pyrin domain-containing 3 (NLRP3) inflammasome which induces pyroptosis. (**C**) Cytosolic ORF3a can activate NLRP3 by the promotion of TNFR-associated factor 3 (TRAF3) -mediated ubiquitination of apoptosis-associated speck-like protein containing a CARD (ASC), enhancing the pyroptosis pathway. (**D**) Crosstalk between the main cell death pathways that occurs during SARS-CoV-2 infection leads to the phenomenon of PANoptosis.

**Table 1 cells-10-01585-t001:** Molecular components of the PANoptosome induced by TNF and IFN-γ stimulus after SARS-Cov-2 infection.

Death Pathways Involved	PANoptosome Component
Pyroptosis	Caspase-1 NLRP3 ASC
Necroptosis	RIPK1 RIPK3
Apoptosis	Caspase-8 FADD

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
