# Peer review of "Cell Death in Coronavirus Infections: Uncovering Its Role during COVID-19"

_cells, 2021, doi:10.3390/cells10071585_

Round 1

Reviewer 1 Report

In this review, the authors included apart from some general characteristics of SARS-CoV-2, immunopathogenic mechanisms present in COVID-19, as well as the major pathways of cell death.

There are a few points that need to be addressed in order to make the manuscript suitable for publication. These are the following:

1) I think that the structure of the review should be reconsidered. More specifically, in part 3 of the text (cell death during viral infections), the 3 different mechanisms are analysed in paragraphs 3.1, 3.2, and 3.3. The same should apply for part 4 (cell death during SARS-CoV-2 infection).  Text should include findings on apoptosis (intrinsic/extrinsic pathways), necroptosis and pyroptosis separately as it would be easier to follow.

In general, part 3 should be more thorough before moving on to the ‘strategies’ part.

2) Some molecules participating in apoptosis are not mentioned, although they have been studied in COVID 19, for example CD95.

3) In pages 13-15 a more logical flow is needed in terms of findings stated. The first paragraphs describe the role of ORF3a (which has not been mentioned before) and of the inflammasome. Then, lymphopenia, IRF1, and cytochrome c are mentioned, but not in an organized way.

4) It would be helpful to include a table with the molecules of PANoptosis found to be related with SARS-CoV-2 infection.

5) The Figure along with the figure legend should be self-explanatory; therefore abbreviations should be explained again.

6) Some paragraphs seem to lack connection with the others. An example is the last paragraph of page 5 about galectins (and not galactins) and the following paragraph on page 6 about immune response in elderly patients. In particular, ii is not very clear if findings described in this paragraph (page 6, paragraph 2) have to do only with elderly patients.

7) In page 15, paragraph 2, line 11, should be rephrased ‘this suggests that…could contribute..’

Author Response

Reviewer#1(Remarks to the Author):

1) I think that the structure of the review should be reconsidered. More specifically, in part 3 of the text (cell death during viral infections), the 3 different mechanisms are analysed in paragraphs 3.1, 3.2, and 3.3. The same should apply for part 4 (cell death during SARS-CoV-2 infection).  Text should include findings on apoptosis (intrinsic/extrinsic pathways), necroptosis and pyroptosis separately as it would be easier to follow. In general, part 3 should be more thorough before moving on to the ‘strategies’ part.
Thank you for the suggestion that allowed us to reconsider the structure of the last paragraphs. We subdivided paragraph 4 in 4.1, 4.2 and 4.3 following the structure of paragraph 3. We also added new findings about the different cell death mechanisms.

2) Some molecules participating in apoptosis are not mentioned, although they have been studied in COVID 19, for example CD95.
Thank you for the useful comment. We amended as suggested and added more information about CD95. The modified text reports “There is also evidence of involvement of CD95 in the induction of apoptosis in lymphoid cells of SARS-CoV-2 patients. CD95 expression increased in circulating CD4+ and CD8+ lymphocytes of infected subjects compared to healthy controls, and a direct correlation between higher CD95 expression and lower CD4+ absolute count was observed. A similar pattern was detected  among CD8+ T cells [107]”.

3) In pages 13-15 a more logical flow is needed in terms of findings stated. The first paragraphs describe the role of ORF3a (which has not been mentioned before) and of the inflammasome. Then, lymphopenia, IRF1, and cytochrome c are mentioned, but not in an organized way.

Thank you for this comment. We added information and references to make the mentioned paragraphs clearer and more organized. Regarding ORF3a, we modified accordingly “The role of Open Reading Frame 3a (ORF3a),  a conserved coronaviruses accessory protein, was investigated [100-101]. ORF3a is a viroporin, a transmembrane protein acting as ion channel, whose function has been associated to the capacity of contribute to virus release [102-104]. In fact, in animal models the deletion of this protein reduced viral replication [105]. Experiments on various cell lines revealed that ORF3a induces the cleavage and activation of caspase-8 without affecting Bcl-2 [101]. This means that ORF3a induces apoptosis via the extrinsic pathway. Inhibitors of caspase-8 or caspase-9 significantly suppress ORF3a-induced apoptosis. ORF3a is also present in SARS-CoV-2 and shares 73% homology with its SARS-CoV counterpart [101]. Mutations that lead to a cytosolic form of the protein eliminates the capacity of inducing apoptosis only for SARS-CoV-2, whereas the membrane-associated feature is involved but not essential in triggering this pathway for SARS-CoV [101]. This likely suggests that SARS-CoV-2 has a weaker pro-apoptotic activity if compared to SARS-CoV, and that this difference can partially explain the different pathogenicity of these viruses. Indeed, SARS-CoV-2 have been generally believed to be less virulent than SARS-CoV and leads to the generation of asymptomatic subjects which helps in a major spreading of the virus [101].”

Concerning IRF1, we modified accordingly “The activation of the protein kinases RIPK1 and RIPK3 and the subsequent phosphorylation of MLKL have also been observed during TNF and IFN-g treatment, suggesting that necroptosis can be induced (Figure 2). Thanks to a deeper analysis of the signaling pathway induced by these cytokines, it was concluded that “inflammatory cell death” depended on interferon regulatory factor 1 (IRF1) and nitric oxide (NO) axis. NO induces T cells death, and T cells lacking inducible NO synthase (iNOS) have reduced post-activation death [147-148]. Regarding the IRF1 pathway, T cells lacking STAT1 are resistant to activation-induced pathway, confirming a role of IRF1 in T cell death mechanisms  [149]. TNF and IFN-, that are present at high concentrations in plasma of COVID-19 patients, may aggravate lymphopenia through direct killing of lymphocytes [145]. Overall, these data suggest that these cytokines together lead to PANoptosis, the phenomenon whereby pyroptosis, apoptosis, and necroptosis are concurrently engaged in the same cell population [150]. The concept of PANoptosis describes a unique inflammatory programmed cell death based on the extensive crosstalk between the main cell death pathways and regulated by PANoptosome, a single complex that provides a molecular scaffold for the interaction of key components of each pathway. The defined composition of the PANoptosome is not well established and it is likely to depend on the different stimulus that induces its assembly. It can include NLRP3, ASC, caspase-1, which are regulators of pyroptosis, caspase-8 which other than apoptosis, it is also involved in the regulation of the other cell death pathways being recruited by FADD, and kinases RIPK1 and RIPK3, that triggers necroptosis [145, 151-152] (Table 1).”

4) It would be helpful to include a table with the molecules of PANoptosis found to be related with SARS-CoV-2 infection.

We thank the reviewer for this comment that helped us to ameliorate the manuscript. We opted to add a paragraph explaining the concept of PANoptosis as a unified mechanism of inflammatory cell death derived from the crosslinked activation of pyroptosis, apoptosis, and necroptosis during SARS-CoV-2 infection (4.4 PANoptosis). Consequently, we included a table containing the principal molecules activated in PANoptosis that constitute PANoptosome, with reference to the cell death pathway to which they belong (Table 1).

5) The Figure along with the figure legend should be self-explanatory; therefore abbreviations should be explained again.
Thank you for the useful comment. We added the explanations of the abbreviation reported in the figure legend as you kindly suggested.

6) Some paragraphs seem to lack connection with the others. An example is the last paragraph of page 5 about galectins (and not galactins) and the following paragraph on page 6 about immune response in elderly patients. In particular, it is not very clear if findings described in this paragraph (page 6, paragraph 2) have to do only with elderly patients.

We acknowledge the reviewer for this comment. We revised the paragraph to make it clearer. In addition, we specified when findings had to do only with elderly patients or when they were not age-related.

7) In page 15, paragraph 2, line 11, should be rephrased ‘this suggests that…could contribute..’

We rephrased the sentence and added information to better explain the reported findings.

Reviewer 2 Report

The authors wrote a review on cell death in COVID-19 infections which described three pathways of cell death – apoptosis, necroptosis and pyroptosis. While it was a generally well-written review, I do have several comments on the current manuscript.

Comments:

  1. It was mentioned several times that cell death such as apoptosis can be a double edged sword, so perhaps it would be clearer for the readers if the authors could state clearer when it is beneficial or disastrous when cell death takes place.
  2. It was a good read on how various viruses can induce different cell death through multiple pathways. It would be nice to see an overview of mechanisms/pathways for each section of cell death (apoptosis, necroptosis, pyroptosis) in figure(s).

Author Response

OVERALL

The authors wrote a review on cell death in COVID-19 infections which described three pathways of cell death – apoptosis, necroptosis and pyroptosis. While it was a generally well-written review, I do have several comments on the current manuscript.

1) It was mentioned several times that cell death such as apoptosis can be a double edged sword, so perhaps it would be clearer for the readers if the authors could state clearer when it is beneficial or disastrous when cell death takes place.

Thank you for the suggestion that allowed us to deepen the topic. We reported as follows “The dysregulation of apoptosis is a feature of a wide variety of diseases. Uncontrolled apoptosis is found in infertility, immunodeficiency, and chronic degenerative diseases such as human neurodegenerative diseases Parkinson and Alzheimer. Whereas delayed or inhibited apoptosis is a hallmark of cancer and autoimmunity [45]. Apoptosis occurs also during viral infections and normally leads to a beneficial outcome for the infected host, but deleterious for the invading virus whose replication is blocked in the absence of a viable susceptible host cell. However, it is also possible that virally induced apoptosis proves harmful to the host, damaging immune cells that should protect it. For example, as demonstrated by Baloul and colleagues, the increased of FasL production after rabies infection in mice induces migratory T cell death via the extrinsic apoptotic pathway, making the host unable to control the viral invasion in the central nervous system. Whereas, in mice lacking FasL the phenotype of the infection was less severe with a reduced number of CD3+ T cells undergoing apoptosis compared to normal mice [46]. In addition, viruses can take advantage from apoptosis as during lytic infection or at late stages of infection this type of cell death can permit the dissemination of their progeny [47].”.

2) It was a good read on how various viruses can induce different cell death through multiple pathways. It would be nice to see an overview of mechanisms/pathways for each section of cell death (apoptosis, necroptosis, pyroptosis) in figure(s).

We thank the reviewer for this suggestion that helped us in ameliorating the paper. We added Figure 2 to generally explain the mechanisms of apoptosis, necroptosis and pyroptosis. We also specified which viruses cited in the text belong to the reported pathways.

Round 2

Reviewer 1 Report

One last comment has to do with paragraph 7 (conclusions). More specifically, I think that it needs some last corrections (for example: 3rd paragraph, '..included in the current arsenal used to contrast COVID-19' is hard to understand. Also: 4th paragraph, 'in conclusion, a variety of pathways' is correct).